# *Carnosinase-1* Knock-Out Reduces Kidney Fibrosis in Type-1 Diabetic Mice on High Fat Diet

**DOI:** 10.3390/antiox12061270

**Published:** 2023-06-14

**Authors:** Tilman Pfeffer, Charlotte Wetzel, Philip Kirschner, Maria Bartosova, Tanja Poth, Constantin Schwab, Gernot Poschet, Johanna Zemva, Ruben Bulkescher, Ivan Damgov, Christian Thiel, Sven F. Garbade, Kristina Klingbeil, Verena Peters, Claus Peter Schmitt

**Affiliations:** 1Centre for Paediatric and Adolescent Medicine, University of Heidelberg, 69120 Heidelberg, Germany; tilman.pfeffer@med.uni-heidelberg.de (T.P.); charlotte.wetzel@med.uni-heidelberg.de (C.W.); philip.kirschner@med.uni-heidelberg.de (P.K.); maria.bartosova@med.uni-heidelberg.de (M.B.); christian.thiel@med.uni-heidelberg.de (C.T.); sven.garbade@med.uni-heidelberg.de (S.F.G.); kristina.klingbeil@med.uni-heidelberg.de (K.K.); 2Tissue Bank of the German Center for Infection Research (DZIF), Partner Site Heidelberg, Institute of Pathology, Heidelberg University Hospital, 69120 Heidelberg, Germany; 3Center for Model System and Comparative Pathology (CMCP), Institute of Pathology, University Hospital Heidelberg, 69120 Heidelberg, Germany; tanja.poth@med.uni-heidelberg.de; 4Institute of Pathology, University Hospital Heidelberg, 69120 Heidelberg, Germany; constantin.schwab@med.uni-heidelberg.de; 5Centre for Organismal Studies (COS), Metabolomics Core Technology Platform, University of Heidelberg, 69120 Heidelberg, Germany; gernot.poschet@cos.uni-heidelberg.de; 6Internal Medicine I and Clinical Chemistry, University Hospital Heidelberg, 69120 Heidelberg, Germany; johanna.zemva@med.uni-heidelberg.de (J.Z.); ruben.bulkescher@med.uni-heidelberg.de (R.B.); 7Institute of Medical Biometry and Informatics, University of Heidelberg, 69120 Heidelberg, Germany; ivan.damgov@med.uni-heidelberg.de

**Keywords:** anserine, carnosine, diabetic nephropathy, fibrosis

## Abstract

Carnosine and anserine supplementation markedLy reduce diabetic nephropathy in rodents. The mode of nephroprotective action of both dipeptides in diabetes, via local protection or improved systemic glucose homeostasis, is uncertain. Global carnosinase-1 knockout mice (*Cndp1*-KO) and wild-type littermates (WT) on a normal diet (ND) and high fat diet (HFD) (*n* = 10/group), with streptozocin (STZ)-induced type-1 diabetes (*n* = 21–23/group), were studied for 32 weeks. Independent of diet, *Cndp1*-KO mice had 2- to 10-fold higher kidney anserine and carnosine concentrations than WT mice, but otherwise a similar kidney metabolome; heart, liver, muscle and serum anserine and carnosine concentrations were not different. Diabetic *Cndp1*-KO mice did not differ from diabetic WT mice in energy intake, body weight gain, blood glucose, HbA1c, insulin and glucose tolerance with both diets, whereas the diabetes-related increase in kidney advanced glycation end-product and 4-hydroxynonenal concentrations was prevented in the KO mice. Tubular protein accumulation was lower in diabetic ND and HFD *Cndp1*-KO mice, interstitial inflammation and fibrosis were lower in diabetic HFD *Cndp1*-KO mice compared to diabetic WT mice. Fatalities occurred later in diabetic ND *Cndp1*-KO mice versus WT littermates. Independent of systemic glucose homeostasis, increased kidney anserine and carnosine concentrations reduce local glycation and oxidative stress in type-1 diabetic mice, and mitigate interstitial nephropathy in type-1 diabetic mice on HFD.

## 1. Introduction

Diabetic nephropathy (DN) is a major complication in patients with type-1 and type-2 diabetes and the leading cause of end-stage kidney disease. According to numerous studies in rodents, supplementation of the histidine-containing dipeptides carnosine and anserine mitigates diabetic nephropathy, based on a broad range of protective actions [1,2,3]. These include anti-oxidative effects [2,4], 4-hydroxynonenal (4-HNE) quenching [5,6,7], methylglyoxal (MG) polymerization [8] and hydrogen sulfide formation [2,9]. In vitro, carnosine inhibits high-glucose-mediated matrix accumulation by interfering with TGF-β production [10]. Anserine exerts higher antioxidative action than carnosine and activates the intracellular Hsp70/HO-1 defense system under oxidative and glycative stress [11]. Carnosine supplementation preserved diabetic kidney morphology, i.e., glomerular hypertrophy [12] and podocyte number [13,14], inhibited mesangial expansion [15] and affected intraglomerular pressure [16]. In several but not all of these studies, however, glucose homeostasis was also improved with carnosine supplementation [17,18,19,20]. Serum carnosine concentrations correlated with β-cell mass, pointing to a direct role of carnosine in the diabetes-affected pancreas [21,22]. Likewise, anserine treatment in type-2 diabetic mice improved nephropathy and glucose homeostasis [11]. Thus, it is unclear how far the nephroprotective actions of carnosine and anserine were exerted via direct protective effects or indirectly via improving glucose homeostasis.

The promising experimental studies in rodents stimulated first clinical trials, which suggest a significant protective potential of carnosine in humans, even though carnosine and anserine are both rapidLy metabolized by human serum carnosinase 1 (CN1) [23,24]. A randomized controlled, double-blinded clinical trial with a daily dose of 2 g L-carnosine in 15 overweight and obese individuals for 12 weeks demonstrated preservation of insulin sensitivity, insulin secretion and glucose tolerance [25]. A randomized placebo-controlled trial in 90 paediatric patients with diabetic nephropathy over 12 weeks demonstrated a decrease in glycated haemoglobin A1c (HbA1c), urinary albumin and alpha 1 microglobulin excretion, i.e., in markers of glomerular and tubular damage with carnosine treatment [26]. Randomized controlled supplementation of 2 g/day of carnosine in 40 patients with diabetic nephropathy for 12 weeks decreased urinary TGF-β, a biomarker of kidney fibrosis, but not glomerular filtration rate, albuminuria or HbA1c [27]. In a recent observational study in 970 patients with type-2 diabetes and no or mild kidney disease and a mean follow up of 1.2 years, baseline serum CN1 concentrations were associated with glomerular filtration rate (GFR) at baseline and predicted renal function impairment; CN1 activity correlated with HbA1c [28]. Patients with a GFR decline of more than 3 mL/min/1.73 m^2^ had a higher serum CN1 activity. Altogether, these findings point to a significant role of carnosine metabolism in patients with diabetic nephropathy. As in rodents, it remains unclear how far the nephroprotective actions of carnosine were exerted via improved systemic glucose homeostasis or direct protective effects in the kidney.

To answer this question, we studied global *Cndp1*-KO mice, that have two- to nine-fold increased kidney carnosine and anserine concentrations, but unchanged concentrations in all other organs and in serum [29]. This might be explained by the previously demonstrated high glomerular and tubular CN1 expression in murine [30] and human kidneys [31]. This model allows discerning specific effects of kidney carnosine and anserine metabolism from the systemic effects of both dipeptides with a secondary impact on diabetic nephropathy. We established type-1 diabetes in these mice, provided normal and high fat diets and carefully characterized systemic glucose homeostasis, kidney metabolome and impact on diabetic nephropathy.

## 2. Materials and Methods

### 2.1. Animal Experiments

Mice from strain B5;129S5*-Cndp1*tmLX/Mmucd (#032215; MMRRC UC Davis, Davis, UT, USA) were used for experiments. The genotype was verified via PCR as recommended by Davis (https://mmrrc.ucdavis.edu/protocols/032215Geno_Protocol.pdf; accessed on 1 April 2019). Animals were housed in a 12:12 h light to dark cycle at 24 °C in the Interfaculty Biomedical Facility (IBF) at Heidelberg University. Food and water were supplied ad libitum. Type-1 diabetes mellitus was induced in female mice at the age of 10 weeks by six consecutive intraperitoneal (i.p.) injections of 50 mg/kg body weight (BW) streptozocin (Sigma Aldrich, Darmstadt, Germany); non-diabetic controls received injections of sodium-citrate buffer (0.05 mM). After diabetes mellitus manifestation (i.e., blood sugar > 300 mg/dL), mice were kept for 32 weeks. For the normal diet (ND), mice were fed Rod16^®^ (Altromin, Lage, Germany); for the high fat diet (HFD), C1090-60^®^ (free of animal origin, Altromin, Lage, Germany) was used. Blood sugar was checked three times a week and insulin (Lantus, Sanofi-Aventis, Frankfurt, Germany) was injected s.c. according to blood sugar levels (Appendix A). Fasting blood glucose, body weight, energy uptake and liquid intake were measured monthly from week 12 on. Spot urine was measured at week 32 (Figure 1). Energy intake and liquid intake were determined by evaluating the decrease of food cup weight and water bottle weight over 48 to 72 h. HbA1c was measured in EDTA blood samples using the Mouse Haemoglobin A1c (HbA1c) Assay Kit (Crystal Chem Inc., Elk Grove Village, IL, USA) following the manufacturer’s protocol. Blood glucose determination as measured after 5 h of fasting. An Intraperitoneal Insulin Tolerance Test (IPITT) and an Intraperitoneal Glucose Tolerance Test (IPGTT) were performed at experimental week 30 after 5 h of fasting. During the 2 h tests mice were fasted and deprived from water. For IPITT 1 U/kg BW insulin (Insuman Rapid, Sanofi-Aventis, Frankfurt, Germany) and for IPGTT 2 g/kg BW D-glucose (20% *w/v* in sterile saline solution) were injected i.p. Blood glucose was determined 5, 10, 15, 20, 30, 45, 60, 80, 100 and 120 min after injection. At sacrifice, organs were harvested, formalin-fixed and paraffin-embedded for histopathology or cryo-preserved for molecular studies. The studies were approved by the respective authorities (Regierungspräsidium, Karlsruhe, Germany, 35-9165 81/G-209/16).

### 2.2. Quantitative Histopathology and Immunohistochemistry

For immunohistochemistry, the kidney was embedded in paraffin after overnight fixation in 4% buffered formalin and routine dehydration. Kidney tissue samples were cut into 4 μm-thick sections, and all tissue sections were stained with hematoxylin and eosin (H&E). Further staining methods included Picrosirius Red, Elastica van Gieson, Constantine’s Picrosirius Red and Periodic acid-Schiff (PAS) staining. Immunohistochemistry was performed on selected kidney samples using the anti-phosphor-NRF2 antibody (diluted 1:50, PA5-105664, ThermoFisher Scientific, Schwerte, Deutschland), anti-AGE (diluted 1:10,000, ab23722; Abcam, Berlin, Deutschland) and anti-TSP-1 Goat IgG (diluted 1:300, AF3074 R&D Systems, Minneapolis, MN, USA). Visualization was performed via Polyview Plus anti-rabbit HRP (ENZ-ACC103-0150, Enzo Life Science, Lörrach, Deutschland) for hosphor-NRF2. For AGE and TSP-1 the anti-Rabbit IgG (whole molecule)-peroxidase antibody (diluted 1:3000, A0545; Sigma-Aldrich, Darmstadt, Deutschland) was used, and visualization was performed with 3,3’-diaminobenzidin (DAB). Stained tissue sections were scanned at an Aperio AT2 (Leica, Wetzlar, Germany) at 40× magnification. Densitometrical analysis of immunohistochemical staining was performed using Aperio ImageScope v12.4. from Leica (Leica, Wetzlar, Germany) with a modified positive pixel count macro (Appendix A). Evaluation of vasculopathy was performed on Elastica van Gieson stained sections by two individual researchers [32]. A minimum of four arterioles (diameter of 60–100 µm) per renal tissue section was evaluated for the lumen and vessel diameter and intima and media thickness on both sides at two opposing sites of the arterioles. Histopathological evaluation of the kidney sections was performed by an experienced veterinary pathologist and a human pathologist with expertise in nephropathology. A semiquantitative scoring system was established comprising diabetes-associated criteria as cystic glomeruli (0 = none, 1 = single, 2 = several, 3 = many glomeruli), mesangial sclerosis, tubular dilation, tubular fibrosis and atrophy, proteinaceous casts, interstitial fibrosis, chronic interstitial inflammation and arterial lesions (0 = none, 1 = mild, 2 = moderate, 3 = severe lesion for each criterion). For arterial lesions, all visually noticeable alterations were considered. The mean of scoring over each group was calculated and used for statistical analysis. Additionally, the following histological findings associated with diabetes mellitus were included in the scoring system by detecting their presence or absence: fibrosis of the Bowman’s capsule, mesangial expansion and glomerular hyalinosis. Scoring was based on the evaluation of the whole tissue sections. All tissue sections were analyzed in blinded manner concerning genotype and the treatments of the mice.

### 2.3. Urine Albumin Quantification

Albumin was measured as mg/l in spot urine samples from experimental week 32 using Mouse Albumin ELISA (ALPCO, Salem, NH, USA). Creatinine determination was performed as g/L with the Creatinine Assay kit (colorimetric) from Abcam (Abcam, Cambridge, UK) with previously centrifuged (10 min at 12,000× *g*) and 1:25 diluted samples (in ddH2O).

### 2.4. Tissue RT-qPCR mRNA and Western Blotting

For mRNA expression analysis, total RNA was extracted from tissue samples using a total RNA Kit from peqGOLD VWR (VWR, Radnor, PA, USA). Extracted mRNA was transcribed to cDNA using SuperScript IV Reverse Transcriptase (ThermoFisher Scientific, Waltham, MA, USA) and gene expression was determined via RT-qPCR with SYBR^®^ Green JumpStart™ Taq ReadyMix™ with gene-specific oligonucleotides (Appendix A). Relative mRNA expression levels of specific genes were analyzed and calculated using the 2-ddCT method. Proteins were quantified by Western immunoblotting. Tissue samples were lysed with RIPA buffer, and 35–40 µg protein lysate was separated via SDS gel-electrophoresis with an 8% SDS gel and transferred on PVDF membrane. For detection of Carnosine synthase 1, the antibody was diluted at 1:500 in 5% milk (Anti-CARNS1, Atlas Antibodies, Bromma, Sweden, HBA0358569). For detection of Hsf1, the specific antibody was diluted at 1:1000 (Cell Signaling Technology, Danvers, MA, USA, #4356), the anti-HSP60 antibody was diluted at 1:1000 (Cell Signaling Technology, Danvers, MA, USA, #12165), the anti-HSPA1a/a1b antibody was diluted at 1:5000 (ProteinTech, Manchester, UK, #100995-1-AP) and the anti-HSPA9 antibody was diluted at 1:500 (ThermoFisher Scientific, Waltham, MA, USA, #MA3-028). Protein expression of the target protein was normalized to the β-actin abundance of the representative sample.

### 2.5. Tissue AGE and 4-HNE Quantification

4-hydroxynonenal (4-HNE) was determined in kidney lysate using OxiSelect™ 4-HNE Adduct Competitive Elisa Kit (Cell Biolabs, Inc., San Diego, CA, USA), and samples were diluted prior to use 1:1 in 1×PBS. Advanced glycation end products were determined in formalin-fixated paraffin-embedded kidney tissue sections using a 1:10,000 dilution of AGE antibody (ab23722, Abcam, Germany). Densitometrical evaluation was performed using Aperio ImageScope v12.4. from Leica (Leica, Wetzlar, Germany) with modified positive pixel count macro (Appendix A).

### 2.6. Tissue Metabolomics

Anserine and carnosine were derivatized by carbozole-9-carbonyl chloride after dilution with sulfosalicylic acid and determined via HPLC as described previously [29]. Amino acids in renal tissue lysates were measured as described [33]. MxP^®^ Quant 500 kit (Biocrates, Innsbruck, Austria)-based metabolomics was used to analyze multiple chemical classes of metabolites. Analysis was performed from frozen kidney tissue samples as described by Andresen et al. after MTBE/EtOH extraction [34].

### 2.7. Statistical Analysis

For all analyses, at least four individual animals per group were used. All data are presented as mean ± standard deviation (SD), except for area under the curve (AUC) analysis (mean ± standard error of the mean (SEM)). Student’s *t*-test or multiple unpaired Student’s *t*-test were used for statistical comparisons of two groups (statistical evaluation of amino acids and dipeptides), and one-way ANOVA with the Tukey test was performed for three or more groups. For statistical evaluation of organ weight/body weight, a two-way ANOVA Tukey test was used. Outliers were identified using the robust regression and outlier removal (ROUT) method with Q = 1% and were removed consequently. Probability of survival was analyzed using the log-rank (Mantel-Cox) test. Analysis of IPITT an IPGTT was performed using the generalized additive model (GAM) with the R package ‘mgcv’ [35]. Statistical analyses were performed using GraphPad Prism 9 software and R.

## 3. Results

### 3.1. Body and Organ Weight

Mean body weight was similar in diabetic *Cndp1*-KO and diabetic WT mice on ND and in respective diabetic mice on HFD. As reported previously, mean body weight was higher in non-diabetic *Cndp1*-KO mice than in non-diabetic WT mice on ND (13.7 ± 7.0%, *p* = 0.017), but not in respective mice on HFD (Appendix A). Energy and liquid intake were not different between genotypes in non-diabetic and diabetic mice on ND and HFD (Appendix A), and organ weights did not differ (Appendix A).

### 3.2. Glucose Homeostasis

Since carnosine has previously been shown to improve glucose homeostasis, we examined the effects of increased kidney carnosine concentrations on glucose homeostasis. Fasting blood glucose (measured seven times from week 12 on), HbA1c (week 32) and insulin requirement (measured seven times from week 12 on) were not different in *Cndp1*-KO mice compared to WT littermates on ND and HFD, respectively (Appendix A). Intraperitoneal insulin and glucose tolerance (week 30) were not different between *Cndp1*-KO and WT mice with diabetes and normal glucose homeostasis, on ND and HFD, respectively (Appendix A).

### 3.3. Tissue Anserine and Carnosine

Kidney anserine and carnosine concentrations were 2- to 10-fold higher in non-diabetic and diabetic *Cndp1*-KO mice under ND and HFD compared to their respective WT littermates (Figure 2, Table 1). By contrast, the genotype did not impact heart, liver, muscle and serum anserine and carnosine concentrations. Serum carnosine concentrations were two- to four-fold higher than serum anserine concentrations in all experimental groups (Table 1). In WT mice, carnosinase-1 activity was below the detection limit in all tissues except for the kidney, and in *Cndp1*-KO mice in all tissues including the kidney [31]. Kidney carnosine-synthase 1 (CS) protein was in the same range for all *Cndp1*-KO and WT mice, except for a two-fold higher CS abundance in diabetic *Cndp1*-KO HFD mice compared to respective WT littermates (Appendix A).

### 3.4. Kidney Metabolome

To assess metabolic effects of the *Cndp1*-KO beyond anserine and carnosine in the kidney, we used the MxP Quant 500 (Biocrates) metabolomics assay that enables the quantitative analysis of up to 630 different metabolites. Quantifications are given in Appendix A. Only a few minor differences in the metabolome of *Cndp1*-KO and WT mice were detected. Diabetic *Cndp1*-KO mice on ND exhibited alterations in the diacylglycerole and triacylglyceride profile compared to diabetic WT on ND. In non-diabetic *Cndp1*-KO mice on HFD, 5 out of 12 diacylglceroles were increased. Amino acid profiles were not altered by genotype, except for an increase in leucine in HFD *Cndp1*-KO vs. HFD WT mice. These findings demonstrate that the metabolic effects of the Cndp1-KO were largely limited to anserine and carnosine.

### 3.5. Kidney Morphology

Histopathological evaluation of kidney tissues revealed a large range of diabetes-associated lesions in *Cndp1*-KO and WT mice on ND and HFD (Table 2), including fibrosis of Bowman’s capsule, glomerular mesangial expansion and sclerosis, tubular protein cylinders, tubular dilation, fibrosis and atrophy and interstitial inflammation and fibrosis. Compared to diabetic WT mice, *Cndp1*-KO mice exhibited less tubular protein accumulation on ND and HFD (Table 2). Diabetic *Cndp1*-KO mice on HFD also had reduced interstitial fibrosis and reduced chronic inflammation (Figure 3). Diabetic mice did not develop diabetic vasculopathy; arteriolar lumen-to-vessel ratio and intima thickness were not different to non-diabetic mice, except for an increase in arteriolar media thickness in diabetic WT HFD mice (Appendix A); the impact of *Cndp1*-KO could not be assessed.

### 3.6. Kidney Stress Marker

Kidney advanced glycation end-product (AGE) concentrations were 1.5-fold higher in diabetic WT mice on ND and HFD, while in corresponding diabetic *Cndp1*-KO mice AGE concentrations were similar to respective non-diabetic controls (Figure 4A, Appendix A). 4-hydroxynonenal (4-HNE), a marker of oxidative stressed-induced lipid peroxidation, was four-fold higher in diabetic versus non-diabetic WT mice, but remained low in diabetic *Cndp1*-KO mice (Figure 4B, Appendix A).

We then studied the expression of molecular pathways involved in diabetic kidney disease. Kidney mRNA and protein abundance of the HSP70 proteins HSPA1A/A1B, HSPA9, HSP60 and HSF1 were independent of the genotype in diabetic and non-diabetic mice (Appendix A) as was the mRNA expression of 17 other markers, with the exception of *Nfe2l2*, which was 30% decreased in diabetic HFD *Cndp1*-KO mice compared to respective WT mice (Appendix A).

Diabetic WT STZ mice on ND and HFD had almost four- and three-fold higher albumin excretion rates than the normoglycemic WT controls at week 32, while the genotype had no influence on albumin excretion (Appendix A). Difference in overall survival rate between diabetic ND *Cndp1*-KO and WT mice was not statistically significant at the end of follow-up after 32 weeks (Appendix A, logrank test *p* = 0.31), but the results indicate a tendency for fatalities to occur later in diabetic *Cndp1-*KO mice. Of the 5 deaths in 23 diabetic ND *Cndp1*-KO mice (21.7%), 3 (60%) died after week 28, while of the 7 fatalities in 21 diabetic WT mice (33.3%), 6 (86%) occurred until week 24. In line with this, survival was superior in diabetic *Cndp1*-KO mice until weeks 20 (*p* = 0.03), 24 and 28 (*p* = 0.08/0.08) but not after 32 weeks (*p* = 0.31). In diabetic mice on HFD, survival was similar between *Cndp1*-KO and WT mice.

## 4. Discussion

The nephroprotective action of anserine and carnosine in diabetes has convincingly been demonstrated in numerous experimental studies. First clinical trials suggest nephroprotective actions also in diabetic patients, even though carnosine and anserine are rapidLy metabolized by human serum CN1. At present, it is, however, unclear in how far these nephroprotective actions are exerted via improved glucose homeostasis [16,18,19,20,21,22] or via direct effects of carnosine and anserine in the kidney. In *Cndp1*-KO mice, we for the first time demonstrated local protective actions of carnosine and anserine independent of systemic effects. Normoglycemic and diabetic *Cndp1*-KO mice have selectively, several-fold increased kidney carnosine and anserine concentrations. In type-1 diabetic mice on ND, the high kidney anserine and carnosine concentrations reduced glycative and oxidative stress, and in type-1 diabetic mice on HFD, they reduced oxidative stress, kidney interstitial inflammation and fibrosis.

We thoroughly studied the *Cndp1*-KO model, to rule out systemic effects acting on the kidney. Energy intake, body weight gain, glucose homeostasis, i.e., average fasting blood glucose, HbA1c, insulin requirements and insulin and glucose tolerance did not differ between *Cndp1*-KO and WT mice. Tissue anserine and carnosine concentrations were selectively 2- to 10-fold, increased in the kidneys. Thus, the model is valid to study specific effects of both dipeptides in the kidney. The fact that the *Cndp1*-KO increased tissue anserine and carnosine concentrations only in the kidney of normoglycemic and diabetic mice indicates a kidney-specific metabolic importance of CN1. In line with this, metabolic CN1 activity was only detectable in the kidney but no other organs under non-diabetic [29] and diabetic conditions, in line with previous mRNA expression data [24]. Of note, the *Cndp1* KO-dependent increase in kidney anserine and carnosine concentrations was not accompanied by further consistent metabolic changes in the kidney, suggesting that the observed differences in the kidney are directly related to anserine and carnosine actions and not to secondary metabolic effects. Broad MxP Quant 500-based metabolomics screening in the kidney revealed alterations in diacylglycerole and triacylglycerole concentrations in diabetic KO versus WT mice on ND, and of diacylglyceroles in non-diabetic mice on HFD. The latter is possibly related to the previously described and now reconfirmed higher mean body weight gain in non-diabetic *Cndp1*-KO versus WT mice [29], which is not explained by differences in energy intake. This effect did not persist in diabetic *Cndp1*-KO mice, where body weight gain was similar, independent of the diet.

The STZ-induced type-1 diabetes resulted in marked glomerular and tubular diabetic lesions and increased albuminuria. In the *Cndp1*-KO mice on ND, the high kidney anserine and carnosine concentrations prevented the 1.5-fold increase in AGE and 4-fold increase in 4-HNE concentrations observed in the diabetic WT littermates, i.e., counteracted the tissue glycative and oxidative stress of diabetes. Surprisingly, this did not result in any histological improvements except for a reduced number in tubular protein casts, but glomerular albuminuria was unchanged. By contrast, in diabetic mice on a high fat diet, the *Cndp1*-KO maintained kidney AGE concentrations on a similar level to that in non-diabetic WT mice and largely prevented any 4-HNE deposition and resulted in reduced interstitial inflammation and fibrosis; however, glomerular damage was unchanged.

A previous study in carnosine-supplemented db/db mice demonstrated reduction of kidney CN1 activity close to non-diabetic mice and a 20-fold increase in kidney anserine concentrations [36]. Subsequent studies in vitro demonstrated higher antioxidative actions of anserine than carnosine in tubular and vascular endothelial cells, and in db/db mice following three intravenous anserine injections, 50% lower albuminuria, lower vascular permeability and lower fasting blood glucose concentrations [11]. This altogether suggests that some of the actions previously reported by carnosine supplementation and in the *Cndp1*-KO mice may be exerted via N-methyltransferase-dependent metabolization to anserine and may also explain the profound reduction in 4-HNE in the diabetic *Cndp1-*KO mice.

Our findings demonstrate that much of the previously reported nephroprotective effects of systemic carnosine and anserine supplementation, which were more pronounced and included preservation of glomerular histomorphology and kidney function [12,14,15,17,36], have been achieved indirectly via systemic effects, such as improved glucose homeostasis [19,20] rather than via direct actions of carnosine and anserine in the kidney. The carbonyl-quenching activity of carnosine and anserine depends on the ratio of reactive carbonyls with carnosine and anserine [8]. The 2- to 10-fold increase in carnosine and anserine was sufficient to reduce the concentration of AGE and 4-HNE.

The predominant tubulo-interstitial effects of the increased kidney anserine and carnosine concentrations in the *Cndp1*-KO mice on HFD are in line with findings from a clinical trial, where randomized controlled supplementation of 2 g/day of carnosine in 40 patients with diabetic nephropathy for 12 weeks decreased urinary TGF-β, a biomarker of kidney fibrosis, but not albuminuria or GFR [27]. High serum CN1 activity in humans largely prevents detectable serum carnosine concentrations after oral intake, but is excreted in urine up to five hours after intake, with the transfer of the dipeptides to the kidney possibly occurring via erythrocytes [37].

Our study has several limitations. Despite the long duration of diabetes and combining type-1 diabetes with HFD, the mice did not exhibit vascular disease after 32 weeks. Thus, the model did not allow discerning systemic from local carnosine/anserine effects on diabetic atherosclerotic kidney disease as reported previously [22]. Likewise, cautious interpretation is required with regard to the superior short-term survival of diabetic *Cndp1*-KO mice on ND versus WT mice. None of the longitudinally assessed parameters differed systematically, and the reduced kidney glycative and oxidative stress observed did not result in morphological improvement in mice on ND. To achieve sufficient statistical power, higher sample sizes are required for the survival rate observed. In addition, it should be of great interest to study the impact of reduced glycative and oxidative stress in the diabetic kidney of the *Cndp1*-KO mice on mitochondrial dysfunction, which has been implicated in kidney fibrosis and progression of kidney disease [38].

## 5. Conclusions

In conclusion, we provide a mouse model of diabetic kidney disease, with selective, several-fold increases in kidney anserine and carnosine concentrations and without differences in systemic glucose homeostasis. The increased kidney anserine and carnosine concentrations resulted in less diabetic glycative and oxidative kidney stress, and in mice on HFD in lower interstitial kidney inflammation and fibrosis. Thus, our findings suggest that much of the previously reported marked nephroprotective actions of systemic carnosine and anserine supplementation in diabetic rodents, markedLy exceeding the effects demonstrated here in diabetic *Cndp1*-KO mice, are exerted via improved glucose homeostasis, rather than direct protective effects of both dipeptides within the kidneys.

## Figures and Tables

**Figure 1 antioxidants-12-01270-f001:**
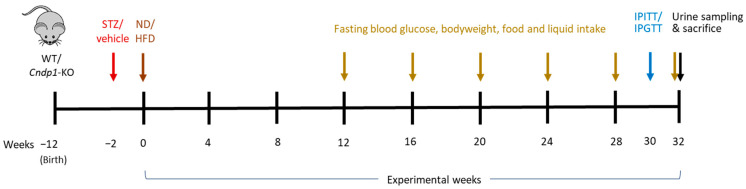
Study scheme. STZ or vehicle was administered to 10-week-old WT and *Cndp1*-KO mice. STZ-treated mice developed diabetes until week 12, when the experiment started. Mice were fed a normal diet (ND) or high fat diet (HFD), resulting in 4 WT mice groups with and without type-1 diabetes on ND and HFD, and 4 respective Cndp1-KO groups. N = 10 in non-diabetic groups and 21–23 in diabetic groups. Insulin was injected according to blood sugar concentrations (measured 3 times a week) in the diabetic mice. IPITT = Intraperitoneal Insulin Tolerance Test; IPGTT = Intraperitoneal Glucose Tolerance Test.

**Figure 2 antioxidants-12-01270-f002:**
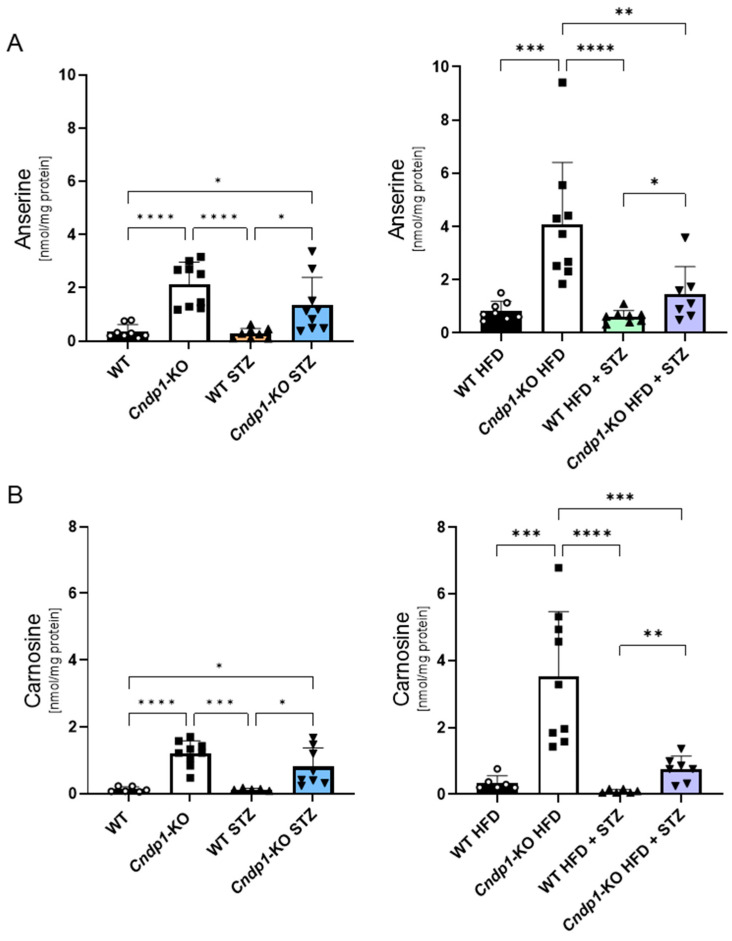
Kidney anserine and carnosine concentrations in diabetic *Cndp1*-KO and WT mice. Kidney anserine (**A**) and carnosine concentrations (**B**) in *Cndp1*-KO and WT mice with diabetes type-1 on normal diet (ND, left panel) and high fat diet (HFD, right panel). *Cndp1*-KO increased kidney anserine and carnosine concentrations 2- to 10-fold compared to their respective WT littermates. Circles depict WT; squares depict *Cndp1*-KO; triangles up depict diabetic WT; triangles down depict diabetic *Cndp1*-KO mice on ND and HFD. Data are mean ± SD, ** p* < 0.05, ** *p* < 0.01, *** *p* < 0.001,**** *p* < 0.0001 (one-way ANOVA followed by Tukey test or unpaired Student’s *t*-test; *n* = 6–9/group).

**Figure 3 antioxidants-12-01270-f003:**
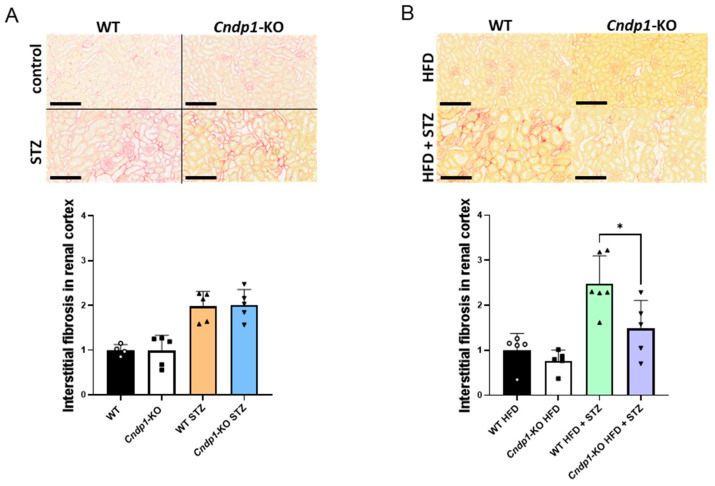
Interstitial fibrosis in diabetic WT and *Cndp1*-KO mice (normal and high fat diet). Interstitial fibrosis (Picrosirius red) in diabetic WT and *Cndp1*-Ko mice on ND (**A**) and HFD (**B**). Statistical comparisons are depicted for the comparison of *Cndp1*-KO mice versus respective WT mice. Circles depict WT; squares depict *Cndp1*-KO; triangles up depict diabetic WT; triangles down depict diabetic *Cndp1*-KO mice on ND and HFD.. Scale bar = 200 µm. Displayed is mean ± SD, * *p* < 0.05.

**Figure 4 antioxidants-12-01270-f004:**
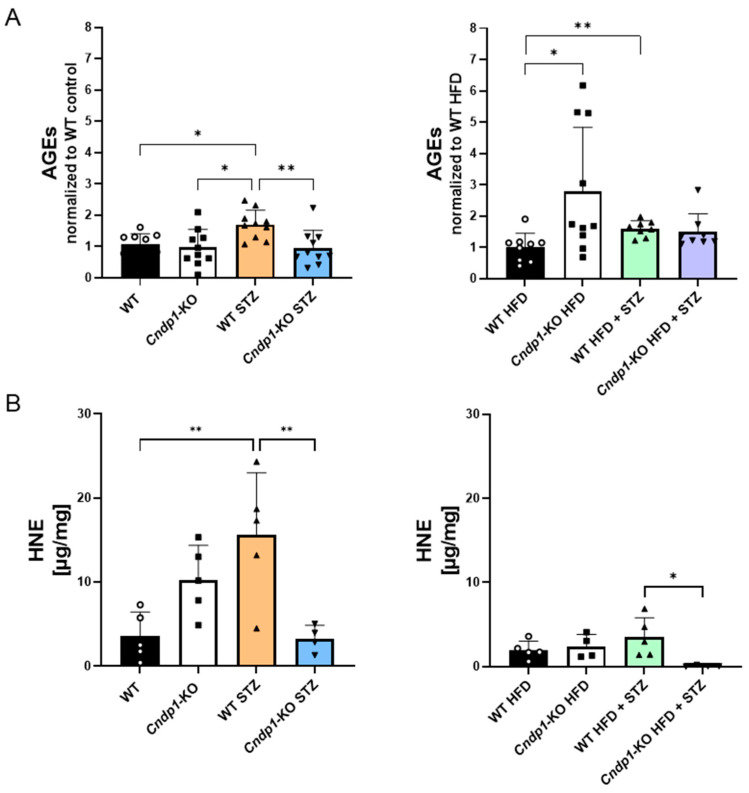
Advanced Glycation Endproducts (AGEs) and 4-hydroxynonenal (4-HNE) in diabetic WT and Cndp1-KO mice (normal and high fat diets). Diabetes-induced increase of AGEs in renal cortex in diabetic mice was lower in diabetic Cndp1-KO mice on ND compared to diabetic WT mice (densitometric evaluation of IHC staining) (**A**). Diabetes-induced increase of 4-HNE was lower in Cndp1-KO mice on ND and on HFD compared to respective littermates (**B**) Circles depict WT; squares depict *Cndp1*-KO; triangles up depict diabetic WT; triangles down depict diabetic *Cndp1*-KO mice on ND and HFD. Data are mean ± SD. * *p* ≤ 0.05, ** *p* ≤ 0.01 (one-way ANOVA with Tukey test and unpaired Student’s *t*-test for AGEs WT HFD vs. WT HFD + STZ).

**Table 1 antioxidants-12-01270-t001:** Anserine and carnosine concentrations in serum and tissue from diabetic and non-diabetic WT and Cndp1-KO mice on normal and high fat diets.

Normal Diet		Non-Diabetic				Diabetic				
	nmol/mg Protein	WT	*Cndp1*-KO	*p* (vs. WT)		WT STZ	*p* (vs. WT)	*Cndp1*-KO STZ	*p* (vs. WT STZ)	
**Kidney ^#^**	Anserine	0.36 ± 0.26	2.13 ± 0.83	<0.001 *		0.28 ± 0.19	0.996	1.35 ± 1.05	0.02 *	
	Carnosine	0.12 ± 0.08	1.20 ± 0.38	<0.001 *		0.12 ± 0.03	>0.999	0.80 ± 0.57	0.021 *	
**Heart**	Anserine	0.13 ± 0.09	0.11 ± 0.07	0.914		0.13 ± 0.04	0.913	0.22 ± 0.06	0.075	
	Carnosine	0.18 ± 0.11	0.15 ± 0.09	0.914		0.32 ± 0.19	0.336	0.57 ± 0.28	0.135	
**Muscle**	Anserine	22.7 ± 8.0	18.2 ± 8.8	0.577		15.3 ± 8.8	0.265	17.0 ± 4.9	0.689	
	Carnosine	9.73 ± 3.82	7.71 ± 3.64	0.577		6.86 ± 3.64	0.265	8.08 ± 2.02	0.689	
**Liver**	Anserine	0.08 ± 0.02	0.08 ± 0.02	0.884		n.a.	n.a.	n.a.	n.a.	
	Carnosine	n.a.	n.a.	n.a.		n.a.	n.a.	n.a.	n.a.	
**Serum**	Anserine	0.01 ± 0.01	0.02 ± 0.01	0.261		0.01 ± 0.00	0.416	0.02 ± 0.02	0.255	
	Carnosine	0.04 ± 0.01	0.05 ± 0.04	0.53		0.02 ± 0.01	0.126	0.06 ± 0.05	0.255	
**High Fat Diet**		**Non-Diabetic**				**Diabetic**				
	**nmol/mg Protein**	**WT HFD**	***Cndp1-*KO HFD**	** *p* ** **(vs. WT HFD)**	** *p* ** **(vs. *Cndp1*-KO)**	**WT** **HFD STZ**	** *p* ** **(vs. WT HFD)**	***Cndp1-*KO HFD STZ**	** *p* ** **(vs. WT STZ HFD)**	** *p* ** **(vs. *Cndp1*-KO STZ)**
**Kidney ^#^**	Anserine	0.84 ± 0.36	4.08 ± 2.33	0.001 *	0.007 *	0.61 ± 0.24	0.987	1.4 ± 1.1	0.046 * ##	>0.999
	Carnosine	0.34 ± 0.22	3.52 ± 1.95	0.001 *	<0.001 *	0.09 ± 0.05	0.981	0.8 ± 0.38	0.002 * ##	>0.999
**Heart**	Anserine	0.10 ± 0.02	0.13 ± 0.04	0.433	0.891	0.13 ± 0.04	0.368	0.10 ± 0.05	0.301	0.016 *
	Carnosine	0.17 ± 0.08	0.15 ± 0.04	0.774	0.966	0.23 ± 0.10	0.368	0.14 ± 0.05	0.301	0.021 *
**Muscle**	Anserine	20.4 ± 13.3	18.5 ± 5.8	0.936	0.999	14.9 ± 5.2	0.597	19.4 ± 8.2	0.481	0.799
	Carnosine	8.13 ± 5.04	7.7 ± 2.3	0.936	0.999	6.1 ± 2.1	0.597	7.9 ± 3.6	0.481	0.895
**Liver**	Anserine	0.08 ± 0.01	0.09 ± 0.04	0.814	0.542	n.a.	n.a.	n.a.	n.a.	n.a.
	Carnosine	n.a.	n.a.	n.a.	n.a.	n.a.	n.a.	n.a.	n.a.	n.a.
**Serum**	Anserine	0.01 ± 0.01	0.02 ± 0.01	0.204	0.847	0.02 ± 0.02	0.578	0.02 ± 0.02	0.702	0.797
	Carnosine	0.04 ± 0.03	0.04 ± 0.02	0.795	0.847	0.06 ± 0.03	0.449	0.04 ± 0.02	0.534	0.733

Statistical analyses were performed with multiple *t*-test, # = one-way ANOVA, ## = Student´s *t*-test. *n* = 4–9 per group. Data are mean ± SD. n.a. = not assessed, *p* values < 0.05 are marked with *.

**Table 2 antioxidants-12-01270-t002:** Kidney histopathology in diabetic and non-diabetic WT and Cndp1-KO mice on normal and high fat diet.

Normal Diet	Non-Diabetic			Diabetic			
	WT	*Cndp1-*KO	*p* (vs. WT)	WT STZ	*p* (vs. WT)	*Cndp1-*KO STZ	*p* (vs. WT STZ)
Diabetic lesions	0.2 ± 0.45	0 ± 0	0.882	1 ± 0	0.019 *	1 ± 0	0.999
Arterial lesions	0 ± 0	0 ± 0	0.999	0 ± 0	0.999	0 ± 0	0.999
Proteinaceous casts	0.6 ± 0.55	0.5 ± 0.58	0.983	1.75 ± 0.5	0.001 *	1 ± 0	0.032
Chronical interstitial inflammation	0.6 ± 0.55	0.5 ± 0.58	0.456	1.25 ± 0.5	0.082	1.4 ± 0.55	0.946
Tubular dilation	0 ± 0	0.25 ± 0.5	0.794	1 ± 0.82	0.002 *	1.6 ± 0.55	0.125
Tubular fibrosis and atrophy	0 ± 0	0.25 ± 0.5	0.794	2 ± 0	<0.001 *	2 ± 0	0.999
Interstitial fibrosis ^#^	1 ± 0.13	0.99 ± 0.35	0.999	1.98 ± 0.34	0.001 *	2.02 ± 0.35	0.999
Fibrosis of Bowman´s capsule	0 ± 0	0 ± 0	0.999	1 ± 0	0.002 *	0.8 ± 0.45	0.882
Cystic glomeruli	0 ± 0	0 ± 0	0.999	0 ± 0	0.999	0.6 ± 0.9	0.125
Glomerular hyalinosis	0.2 ± 0.45	0 ± 0	0.882	0.25 ± 0.5	0.998	0 ± 0	0.794
Mesangial sclerosis	0.2 ± 0.45	0 ± 0	0.884	1.5 ± 0.58	<0.001 *	1.4 ± 0.55	0.983
Mesangial expansion ^#^	0.39 ± 0.02	0.49 ± 0.09	0.011 *	0.47 ± 0.04	0.039 *	0.5 ± 0.07	0.814
Glomerular size ^#^	0.07 ± 0.01	0.07 ± 0.01	0.823	0.07 ± 0.01	0.998	0.07 ± 0.01	0.789
**High Fat Diet**	**Non-Diabetic**			**Diabetic**			
	**WT HFD**	***Cndp1*-KO HFD**	** *p* ** **(vs. WT HFD)**	**WT HFD STZ**	** *p* ** **(vs. WT HFD)**	***Cndp1*-KO HFD + STZ**	** *p* ** **(vs. WT HFD STZ)**
Diabetic lesions	n.a.	n.a.	n.a.	1 ± 0	n.a.	1 ± 0	0.999
Arterial lesions	n.a.	n.a.	n.a.	0 ± 0	n.a.	0 ± 0	0.999
Proteinaceous casts	n.a.	n.a.	n.a.	1.8 ± 0.45	n.a.	1 ± 0	0.004
Chronic interstitial inflammation	n.a.	n.a.	n.a.	2 ± 0.36	n.a.	1 ± 0.71	0.022
Tubular dilation	n.a.	n.a.	n.a.	2.8 ± 0.45	n.a.	2 ± 0.71	0.065
Tubular fibrosis and atrophy	n.a.	n.a.	n.a.	2 ± 0	n.a.	1.6 ± 0.55	0.141
Interstitial fibrosis #	1 ± 0.37	0.77 ± 0.24	0.8732	2.48 ± 0.62	0.001	1.49 ± 0.63	0.020
Fibrosis of Bowman´s capsule	n.a.	n.a.	n.a.	1 ± 0	n.a.	0.8 ± 0.45	0.347
Cystic glomeruli	n.a.	n.a.	n.a.	0.2 ± 0.45	n.a.	0 ± 0	0.347
Glomerular hyalinosis	n.a.	n.a.	n.a.	0 ± 0	n.a.	0.4 ± 0.55	0.141
Mesangial sclerosis	n.a.	n.a.	n.a.	1.2 ± 0.84	n.a.	1.6 ± 0.9	0.486
Mesangial expansion #	0.63 ± 0.06	0.59 ± 0.06	0.6202	0.41 ± 0.05	<0.001	0.44 ± 0.09	0.625
Glomerular size #	0.08 ± 0.01	0.08 ± 0.01	0.9999	0.07 ± 0.01	<0.001	0.07 ± 0.01	0.890

Interstitial fibrosis was determined by densitometrical analysis of Sirius red staining, and mesangial expansion with PAS Staining. Glomerular size was assessed by Aperio image scope^®^. Mesangial area quantification included the glomeruli and the bowman capsular space. n.a. = not assessed; parameters marked with # were analyzed using one-way ANOVA. Two-way ANOVA was used for normal diet and multiple *t*-test for high fat diet, *n* = 4–5 per group, *p* values < 0.05 are marked with *.

## Data Availability

Not applicable.

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
