# Peer review of "Carnosinase-1 Knock-Out Reduces Kidney Fibrosis in Type-1 Diabetic Mice on High Fat Diet"

_antioxidants, 2023, doi:10.3390/antiox12061270_

Round 1

Reviewer 1 Report

antioxidants-2390369

Title: Increased renal anserine and carnosine reduces fibrosis in STZ-induced diabetic Cndp1-KO mice under high fat diet

In this paper, authors investigated the weather carnosine and anserine supplementation may reduce diabetic nephropathy in Global carnosinase-1 knockout mice (Cndp1-KO) and wild-type littermates (WT) on normal 31 diet (ND) and high-fat-diet (HFD), with streptozocin (STZ) induced type 1 diabetes, were studied 32 for 32 weeks.

Minor revision

1.      Authors should be compared the concentration of carnosine and anserine in the blood.

2.      How is expression levels of carnosinase-1 in the kidney, authors may be compared the expression level using immunohistochemistry analysis.

Author Response

Dear Reviewer,

We thank the reviewer for the constructive revision which helped to improve the paper. Please find enclosed our detailed point by point response.

Reviewer 2 Report

In the manuscript “Increased renal anserine and carnosine reduces fibrosis in STZ- induced diabetic Cndp1-KO mice under high fat diet”, the authors proposed to study several parameters of carnosinase-1 knockout mice (Cndp1-KO) and wild-type littermates (WT) on normal diet (ND) and high-fat-diet (HFD), with streptozocin (STZ) induced type 1 diabetes. The paper is of interest and presents some interesting data. However, some drawbacks hamper the enthusiasm. I have some comments that may be useful.

Specific comments:

1. Abstract needs some revisions. For instance, the total number of animals used is not clear. Also, the hypothesis should be clear. Kidney should also be in spotlight on title and abstract. It is unclear the type of tissue reported.

2. Introduction – please provide the rationale for the experiments and the gap in the literature. Perhaps would be good to highlight that the role of mitochondria in pathophysiology of CKD is still debatable (PMID: 35163697) and that the role of Kidney metabolism in fibrosis (cause or consequence?) is still unknown.  

3. Results could be better described, particularly by highlighting some of the most relevant. Sometimes it is useful to numerically describe the results to understand how different are the results and if the difference has any relevance.

4. Please provide a discussion of study limitations. It is also important to explain the new findings and build a take home message.

Acceptable

Author Response

(The authors gave the same response as above.)

Reviewer 3 Report

With all the obvious relevance of the topic of coping with fibrosis, the article suffers greatly from unnecessary redundancy, which is not only necessary but also seriously interferes with the assessment of the quality of work, since navigation within the manuscript and supplementary materials is very difficult and time-consuming distracting from the main goal – how to tackle fibrosis.

The first example is Fig. 1 and Table 1, and it is unclear why to bring them both when a graph is built based on the data of the table. (It is also noticeable that the authors note a strong variability of data for the content of carnosine and anserine in knockout animals on a fat diet. I would be glad to know what the authors think about this – did these animals have any other important distinguishing features that accompanied different values of the response?).

The same redundancy is observed in Figure 3, where numerical data on fibrosis are given in Table.2. By the way, it is completely meaningless to compare bar 2 with bar 3 (white bar vs brown on the left and white bar vs green on the right), where even a high significance of the difference is given (***). Why the comparison of these columns was made, from the point of view of science, is not clear.

At the same time, Figure 2 does not carry explanations at all either in the methodological part or in the figure legends. By the way, it is quite difficult to evaluate the conclusions drawn on the basis of this figure, but if I correctly understood and appreciated the color of different points, then very significant changes occur precisely for triglycerides, while the text states that diglycerides in knockout mice  exposed to different diets differ most significantly.

Finally, if compare Fig 1A and Fig 4A, it is clear that carnosine is not protective against AGE after feeding with high fat, which is not reflected in the manuscript, and instead, it is claimed that both anserine and carnosine reduce glycative stress (line 407)

Minor comment

I guess that in line 218 the authors meant energy intake rather than solely energy which has not been justified

In conclusion, I must say that It seems to me that the authors have "drowned" in the data and really want to "drown" the reader in it as well. I think that in order to be accepted for publication in Antioxidants, the manuscript should be very significantly shortened with the elimination of redundancy and the presentation of only the data that significantly bring us closer to understanding the problem. I note that this data should not copy the already available data on the use of these knockout animals (doi: 10.3390/ijms21144887).

Author Response

(The authors gave the same response as above.)

Reviewer 4 Report

I found the article sent for review interesting. The problem of diabetic nephropathy is essential; still, many issues are not fully understood. The authors conducted their study on Global carnosinase-1 knockout mice (Cndp1-KO) and wild-type littermates on a normal and high-fat diet with streptozocin-induced type 1 diabetes. The research lasted for 32 weeks.

 My request to authors for clarification::
1. I suggest modifying the title. The current one is too literal and contains too many abbreviations, so I don't know if this will affect the article's citation and interest in it in general. I also think "diabetic nephropathy" is missing from the title.
2. There is no precise aim of the study. It should be added.
3. A research diagram would also be helpful - making it easier for the reader to follow the study.
4. Conclusions are too general. They should be clarified.
I am also not sure whether the results of the research conducted by the researchers directly follow the sentence found in the conclusions: "Thus, our findings suggest that much of the previously reported, marked nephroprotective actions of carnosine and anserine supplementation in diabetic rodents is exerted via systemic rather than direct local effects
within the kidneys." What do the authors mean?

5. The arrangement of the literature is incorrect - there are quotations in parentheses in the text instead of reference numbers. The layout of the literature is also in the wrong format.

6. More information on carnosine and anserine would be helpful in the Introduction or Discussion section. Carnosine, for example, inhibits the formation of end products of
advanced glycation of proteins (AGEs).
end products), oxygen free radicals, autocrine
angiotensin II production, and fibronectin synthesis. Similarly - about anserine - in-depth information is missing in the introduction.
This should be supplemented.

Other than that, I have no more complaints. I consider the methodology, the study itself, and the presentation of the results to be impeccable.
There are no limitations to the study. Maybe it could be helpful to add.

English is correct; only minor editing is required.

Author Response

(The authors gave the same response as above.)

Round 2

Reviewer 3 Report

I have to stand by my opinion – in this version of the manuscript, it is absolutely unreadable, and the reader will be forced to drown in hundreds of insignificant data for understanding the essence of the study, and my advice to significantly shorten this manuscript and remove redundancy was fully ignored, moreover, it even increased slightly to some extent (apparently due to the advice of other reviewers). But it seems to me that in this form it will not find readers of Antioxidants.

I also draw the attention to the fact that either the authors, due to their negligence, did not read the version for submission, or something happened in the editorial office, but the line numbers overlaid the figures (Figs 3A and 4A. B) and it became very difficult to catch these figures, which forces to thoroughly compare the first and last versions of the manuscript.
